# Teach Me to Explain: A Review of Datasets for Explainable Natural Language Processing

**Sarah Wiegreffe**[*]
School of Interactive Computing
Georgia Institute of Technology
saw@gatech.edu

**Ana Marasović**[*]
Allen Institute for AI
University of Washington
anam@allenai.org

## Abstract

Explainable Natural Language Processing (EXNLP) has increasingly focused on collecting human-annotated textual explanations. These explanations are used downstream in three ways: as data augmentation to improve performance on a predictive task, as supervision to train models to produce explanations for their predictions, and as a ground-truth to evaluate model-generated explanations. In this review, we identify 65 datasets with three predominant classes of textual explanations (highlights, free-text, and structured), organize the literature on annotating each type, identify strengths and shortcomings of existing collection methodologies, and give recommendations for collecting EXNLP datasets in the future.

## 1 Introduction

Interpreting supervised machine learning (ML) models is crucial for ensuring their reliability and trustworthiness in high-stakes scenarios. Models that produce justifications for their individual predictions (sometimes referred to as *local explanations*) can be inspected for the purposes of debugging, quantifying bias and fairness, understanding model behavior, and ascertaining robustness and privacy [83]. These benefits have led to the development of datasets that contain human justifications for the true label (overviewed in Tables 3–5). In particular, human justifications are used for three goals: (i) to aid models with additional training supervision [142], (ii) to train interpretable models that explain their own predictions [20], and (iii) to evaluate plausibility of model-generated explanations by measuring their agreement with human explanations [29].

Dataset collection is the most under-scrutinized component of the ML pipeline [93]—it is estimated that 92% of ML practitioners encounter data cascades, or downstream problems resulting from poor data quality [109]. It is important to constantly evaluate data collection practices critically and standardize them [13, 39, 95]. We expect that such examinations are particularly valuable when many related datasets are released contemporaneously and independently in a short period of time, as is the case with EXNLP datasets.

This survey aims to review and summarize the literature on collecting textual explanations, highlight what has been learned to date, and give recommendations for future dataset construction. It complements other explainable AI (XAI) surveys and critical retrospectives that focus on definitions, methods, and/or evaluation [33, 15, 77, 1, 103, 51, 42, 133, 26, 44, 82, 121, 12, 86, 54, 19], but not on datasets. We call such datasets EXNLP datasets, because modeling them for the three goals mentioned above requires NLP techniques. Datasets and methods for explaining fact checking [65] and reading comprehension [117] have been reviewed; we are the first to review all datasets with textual explanations regardless of task, comprehensively categorize them into three distinct classes, and provide critical retrospectives and best-practice recommendations.

---

[*] Equal contributions.

35th Conference on Neural Information Processing Systems (NeurIPS 2021) Track on Datasets and Benchmarks.

| Instance | Explanation |
|---|---|
| *Premise:* A white race dog wearing the number eight runs on the track. *Hypothesis:* A white race dog runs around his yard. *Label:* contradiction | `(highlight)` *Premise:* A white race dog wearing the number eight runs on the `track` . *Hypothesis:* A white race dog runs around his `yard` . |
| | `(free-text)` A race track is not usually in someone's yard. |
| *Question:* Who sang the theme song from Russia With Love? *Paragraph:* …The theme song was composed by Lionel Bart of Oliver! fame and sung by Matt Monro… *Answer:* Matt Monro | `(structured)` *Sentence selection:* (not shown) *Referential equality:* "the theme song from russia with love" (from question) = "The theme song" (from paragraph) *Entailment:* X was composed by Lionel Bart of Oliver! fame and sung by ANSWER. ⊢ ANSWER sung X |

Table 1: Examples of explanation types discussed in §2. The first two rows show a highlight and free-text explanation for an E-SNLI instance [20]. The last row shows a (partial) structured explanation from QED for a NATURALQUESTIONS instance [70].

| Instance with Highlight | Highlight Type Clarification |
|---|---|
| *Review:* this film is `extraordinarily horrendous` and I'm not going to waste any more words on it. *Label:* negative | `(¬comprehensive)` *Review:* this film is ▮▮▮▮▮▮▮ and I'm not going to waste any more words on it. |
| *Review:* this film is `extraordinarily horrendous` and I'm not going to `waste any more words on it` . *Label:* negative | `(comprehensive)` *Review:* this film is ▮▮▮▮▮▮ and I'm not going to ▮▮▮▮▮▮▮▮. |
| *Premise:* A shirtless man wearing white shorts. *Hypothesis:* A `man` in white shorts is `running on the sidewalk`. *Label:* neutral | `(¬sufficient)` *Premise:* ▮▮▮▮▮▮▮▮▮▮ *Hypothesis:* ▮ `man` ▮▮▮▮▮▮ `running on the sidewalk`. |

Table 2: Examples of highlights differing in comprehensiveness and sufficiency (discussed in §2, §4).

We first define relevant ExNLP terminology (§2) and overview 65 existing datasets (§3), accompanied with a live version of the tables as a website accepting community contributions: https://exnlpdatasets.github.io. We next analyze what can be learned from existing data collection methodologies. In §4 and §5, we highlight two points that we expect to be particularly important to the current ExNLP research. Specifically, §4 discusses the traditional process of collecting explanations by asking annotators to highlight parts of the input, and its discrepancies with evaluating model-generated highlight explanations. We also draw attention to how assumptions made for collecting free-text explanations (introduced in §2) influence their modeling, and call for better documentation of explanation collection. In §5, we illustrate that not all template-like free-text explanations are incorrect, and call for embracing the structure of an explanation when appropriate. Unlike discussions in §4–5 that are motivated by ExNLP modeling and evaluation choices, the rest of this paper reflects on relevant points from a broader NLP research. In §6, we present a proposal for controlling quality in explanation collection, and in §7, gather recommendations from related subfields to further reduce data artifacts by increasing diversity of collected explanations.

## 2 Explainability Lexicon

An explanation can be described as a "three-place predicate: *someone* explains *something* to *someone*" [50]. The *something* being explained in machine learning systems are task labels: explanations are implicitly or explicitly designed to answer the question "why is [input] assigned [label]?". However, collected explanations can vary in format. We identify three types in the ExNLP literature: *highlights*, *free-text*, and *structured* explanations. An example of each type is given in Table 1. Since a consensus on terminology has not yet been reached, we describe each type below.

**Highlights** are subsets of the input elements (words, phrases, or sentences) that explain a prediction. Lei et al. [73] coin them *extractive rationales*, or subsets of the input tokens of a textual task that satisfy two properties: (i) *compactness*, they are short and coherent, and (ii) *sufficiency*, they suffice for prediction as a substitute of the original text. Yu et al. [141] introduce a third criterion, (iii) *comprehensiveness*, that all the evidence that supports the prediction is selected, not just a sufficient set. Since the term "rationale" implies human-like intent, Jacovi and Goldberg [55] argue to call this type of explanation *highlights* to avoid inaccurately attributing human-like social behavior to AI systems. They are also called *evidence* in fact-checking and multi-document question answering (QA) [65]—a part of the source that refutes/supports the claim. To reiterate, highlights should be sufficient to explain a prediction and compact; if they are also comprehensive, we call them *comprehensive*

| Dataset | Task | Granularity | Collection | # Instances |
|---|---|---|---|---|
| MOVIEREVIEWS [142] | sentiment classification | none | author | 1,800 |
| MOVIEREVIEWS$_c$ [29] | sentiment classification | none | crowd | 200$^{‡◇}$ |
| SST [113] | sentiment classification | none | crowd | 11,855$^◇$ |
| WIKIQA [136] | open-domain QA | sentence | crowd + authors | 1,473 |
| WIKIATTACK [22] | detecting personal attacks | none | students | 1089$^◇$ |
| E-SNLI$^†$ [20] | natural language inference | none | crowd | ∼569K (1 or 3) |
| MULTIRC [60] | reading comprehension QA | sentences | crowd | 5,825 |
| FEVER [118] | verifying claims from text | sentences | crowd | ∼136K$^‡$ |
| HOTPOTQA [137] | reading comprehension QA | sentences | crowd | 112,779 |
| Hanselowski et al. [47] | verifying claims from text | sentences | crowd | 6,422 (varies) |
| NATURALQUESTIONS [68] | reading comprehension QA | 1 paragraph | crowd | n/a$^‡$ (1 or 5) |
| CoQA [104] | conversational QA | none | crowd | ∼127K (1 or 3) |
| COS-E v1.0$^†$ [100] | commonsense QA | none | crowd | 8,560 |
| COS-E v1.11$^†$ [100] | commonsense QA | none | crowd | 10,962 |
| BOOLQ$_c$ [29] | reading comprehension QA | none | crowd | 199$^{‡◇}$ |
| EVIDENCEINFERENCE v1.0 [71] | evidence inference | none | experts | 10,137 |
| EVIDENCEINFERENCE v1.0$_c$ [29] | evidence inference | none | experts | 125$^‡$ |
| EVIDENCEINFERENCE v2.0 [30] | evidence inference | none | experts | 2,503 |
| SCIFACT [123] | verifying claims from text | 1-3 sentences | experts | 995$^‡$ (1-3) |
| Kutlu et al. [67] | webpage relevance ranking | 2-3 sentences | crowd | 700 (15) |
| SCAT [139] | document-level machine translation | none | experts | ∼14K |
| ECTHR [24] | alleged legal violation prediction | paragraphs | auto + expert | ∼11K |
| HUMMINGBIRD [48] | style classification | words | crowd | 500 |
| HATEXPLAIN [79] | hate-speech classification | phrases | crowd | 20,148 (3) |

Table 3: Overview of datasets with textual **highlights**. Values in parentheses indicate number of explanations collected per instance (if > 1). DeYoung et al. [29] collected or recollected annotations for prior datasets (marked with the subscript $c$). ◇ Collected > 1 explanation per instance but only release 1. † Also contains free-text explanations. ‡ A subset of the original dataset that is annotated. It is not reported what subset of NATURALQUESTIONS has both a long and short answer.

*highlights*. Although the community has settled on criteria (i)–(iii) for highlights, the extent to which collected datasets (Table 3) reflect them varies greatly, as we will discuss in §4. Table 2 gives examples of sufficient vs. non-sufficient and comprehensive vs. non-comprehensive highlights.

**Free-text explanations** are free-form textual justifications that are not constrained to the words or modality of the input instance. They are thus more expressive and generally more readable than highlights. This makes them useful for explaining reasoning tasks where explanations must contain information outside the given input sentence or document [20, 128]. They are also called *textual* [62] or *natural language explanations* [20], terms that have been overloaded [98]. Synonyms, *free-form* [20] or *abstractive explanations* [87] do not emphasize that the explanation is textual.

Finally, **structured explanations** are explanations that are not entirely free-form although they are still written in natural language. For example, there may be constraints placed on the explanation-writing process, such as the required use of specific inference rules. We discuss the recent emergence of structured explanations in §5. Structured explanations do not have one common definition; we elaborate on dataset-specific designs in §3. An example is given in the bottom row of Table 1.

## 3 Overview of Existing Datasets

We overview currently available EXNLP datasets by explanation type: highlights (Table 3), free-text explanations (Table 4), and structured explanations (Table 5). Besides SCAT [139], to the best of our knowledge, all existing datasets are in English. The authors of ∼66% papers cited in Tables 3–5 report the dataset license in the paper or a repository, and 45.61% use *common* permissive licenses; for more information see Appendix B. See Appendix C for collection details.

For each dataset, we report the number of instances (input-label pairs) and the number of explanations per instance (if > 1). The annotation procedure used to collect each dataset is reported as: crowd-annotated ("crowd"); automatically annotated through a web-scrape, database crawl, or merge of existing datasets ("auto"); or annotated by others ("experts", "students", or "authors"). Some authors perform semantic parsing on collected explanations (denoted with ∗); we classify them by the dataset type before parsing and list the collection type as "crow + authors". Tables 3–5 elucidate that the dominant collection paradigm (≥90%) is via human (crowd, student, author, or expert) annotation.

| Dataset | Task | Collection | # Instances |
|---|---|---|---|
| Jansen et al. [56] | science exam QA | authors | 363 |
| Ling et al. [76] | solving algebraic word problems | auto + crowd | ∼101K |
| Srivastava et al. [115]* | detecting phishing emails | crowd + authors | 7 (30-35) |
| BABBLELABBLE [46]* | relation extraction | students + authors | 200[‡‡] |
| E-SNLI [20] | natural language inference | crowd | ∼569K (1 or 3) |
| LIAR-PLUS [4] | verifying claims from text | auto | 12,836 |
| COS-E v1.0 [100] | commonsense QA | crowd | 8,560 |
| COS-E v1.11 [100] | commonsense QA | crowd | 10,962 |
| ECQA [2] | commonsense QA | crowd | 10,962 |
| SEN-MAKING [124] | commonsense validation | students + authors | 2,021 |
| CHANGEMYVIEW [10] | argument persuasiveness | crowd | 37,718 |
| WINOWHY [144] | pronoun coreference resolution | crowd | 273 (5) |
| SBIC [111] | social bias inference | crowd | 48,923 (1-3) |
| PUBHEALTH [64] | verifying claims from text | auto | 11,832 |
| Wang et al. [125]* | relation extraction | crowd + authors | 373 |
| Wang et al. [125]* | sentiment classification | crowd + authors | 85 |
| E-δ-NLI [18] | defeasible natural language inference | auto | 92,298 (∼8) |
| BDD-X[††] [62] | vehicle control for self-driving cars | crowd | ∼26K |
| VQA-E[††] [75] | visual QA | auto | ∼270K |
| VQA-X[††] [94] | visual QA | crowd | 28,180 (1 or 3) |
| ACT-X[††] [94] | activity recognition | crowd | 18,030 (3) |
| Ehsan et al. [34][††] | playing arcade games | crowd | 2000 |
| VCR[††] [143] | visual commonsense reasoning | crowd | ∼290K |
| E-SNLI-VE[††] [32] | visual-textual entailment | crowd | 11,335 (3)[‡] |
| ESPRIT[††] [101] | reasoning about qualitative physics | crowd | 2441 (2) |
| VLEP[††] [72] | future event prediction | auto + crowd | 28,726 |
| EMU[††] [27] | reasoning about manipulated images | crowd | 48K |

Table 4: Overview of EXNLP datasets with **free-text explanations** for textual and visual-textual tasks (marked with †† and placed in the lower part). Values in parentheses indicate number of explanations collected per instance (if > 1). ‡ A subset of the original dataset that is annotated. ‡‡ Subset publicly available. ∗ Authors semantically parse the collected explanations.

**Highlights (Table 3)** The granularity of highlights depends on the task they are collected for. The majority of authors do not place a restriction on granularity, allowing words, phrases, or sentences of the original input document to be selected. The coarsest granularity in Table 3 is one or more paragraphs in a longer document [68, 24]. We exclude datasets that include an associated document as evidence without specifying the location of the explanation within the document (namely document retrieval datasets). We exclude BEERADVOCATE [80] because it has been retracted.

Some highlights are re-purposed from annotations for a different task. For example, MULTIRC [60] contains sentence-level highlights that indicate justifications of answers to questions. However, they were originally collected for the authors to assess that each question in the dataset requires multi-sentence reasoning to answer. Another example is STANFORD SENTIMENT TREEBANK [SST; 113] which contains crowdsourced sentiment annotations for word phrases extracted from movie reviews [90]. Word phrases that have the same sentiment label as the review can be heuristically merged to get phrase-level highlights [23]. Other highlights in Table 3 are collected by instructing annotators. Instead of giving these instructions verbatim, their authors typically describe them concisely, e.g., they say annotators are asked to highlight words justifying, constituting, indicating, supporting, or determining the label, or words that are essential, useful, or relevant for the label. The difference in wording of these instructions affects how people annotate explanations. In §4, we discuss how one difference in annotation instructions (requiring comprehensiveness or not) can be important.

**Free-Text Explanations (Table 4)** This is a popular explanation type for both textual and visual-textual tasks, shown in the first and second half of the table, respectively. Most free-text explanations are generally no more than a few sentences per instance. One exception is LIAR-PLUS [5], which contains the conclusion paragraphs of web-scraped human-written fact-checking summaries.

**Structured Explanations (Table 5)** Structured explanations take on dataset-specific forms. One common approach is to construct a chain of facts that detail the reasoning steps to reach an answer

| Dataset | Task | Explanation Type | Collection | # Instances |
|---|---|---|---|---|
| WORLDTREE V1 [57] | science exam QA | explanation graphs | authors | 1,680 |
| OPENBOOKQA [81] | open-book science QA | 1 fact from WORLDTREE | crowd | 5,957 |
| Yang et al. [135][††] | action recognition | lists of relations + attributes | crowd | 853 |
| WORLDTREE V2 [132] | science exam QA | explanation graphs | experts | 5,100 |
| QED [70] | reading comp. QA | inference rules | authors | 8,991 |
| QASC [61] | science exam QA | 2-fact chain | authors + crowd | 9,980 |
| EQASC [58] | science exam QA | 2-fact chain | auto + crowd | 9,980 ($\sim$10) |
| + PERTURBED | science exam QA | 2-fact chain | auto + crowd | n/a[‡] |
| EOBQA [58] | open-book science QA | 2-fact chain | auto + crowd | n/a[‡] |
| Ye et al. [138][*] | SQUAD QA | semi-structured text | crowd + authors | 164 |
| Ye et al. [138][*] | NATURALQUESTIONS QA | semi-structured text | crowd + authors | 109 |
| R$^4$C [53] | reading comp. QA | chains of facts | crowd | 4,588 (3) |
| STRATEGYQA [41] | implicit reasoning QA | reasoning steps w/ highlights | crowd | 2,780 (3) |
| TRIGGERNER | named entity recognition | groups of highlighted tokens | crowd | $\sim$7K (2) |

Table 5: Overview of EXNLP datasets with **structured explanations** (§5). Values in parentheses indicate number of explanations collected per instance (if > 1). †† Visual-textual dataset. ∗ Authors semantically parse the collected explanations. ‡ Subset of instances annotated with explanations is not reported. Total # of explanations is 855 for EQASC PERTURBED and 998 for EOBQA.

("chains of facts"). Another is to place constraints on the textual explanations that annotators can write, such as requiring the use of certain variables in the input ("semi-structured text").

The WORLDTREE datasets [57, 132] propose explaining elementary-school science questions with a combination of chains of facts and semi-structured text, termed "explanation graphs". The facts are individual sentences written by the authors that are centered around a set of shared relations and properties. Given the chain of facts for an instance (6.3 facts on average), the authors can construct an explanation graph by linking shared words in the question, answer, and explanation.

OPENBOOKQA [OBQA; 81] uses single WORLDTREE facts to prime annotators to write QA pairs. Similarly, each question in QASC [61] contains two associated science facts from a corpus selected by human annotators who wrote the question. Jhamtani and Clark [58] extend OBQA and QASC with two-fact chain explanation annotations, which are automatically extracted from a fact corpus and validated with crowdsourcing. The resulting datasets, EQASC and EOBQA, contain multiple valid and invalid explanations per instance, as well as perturbations for robustness testing (EQASC-PERTURBED).

A number of structured explanation datasets supplement datasets for reading comprehension. Ye et al. [138] collect semi-structured explanations for NATURALQUESTIONS [68] and SQUAD [102]. They require annotators to use phrases in both the input question and context, and limit them to a small set of connecting expressions. Inoue et al. [53] collect R$^4$C, fact chain explanations for HOTPOTQA [137]. Lamm et al. [70] collect explanations for NATURALQUESTIONS that follow a linguistically-motivated form (see the example in Table 1). We discuss structured explanations further in §5.

# 4 Link Between ExNLP Data, Modeling, and Evaluation Assumptions

All three parts of the machine learning pipeline (data collection, modeling, and evaluation) are inextricably linked. In this section, we discuss what EXNLP modeling and evaluation research reveals about the qualities of available EXNLP datasets, and how best to collect such datasets in the future.

Highlights are usually evaluated following two criteria: (i) *plausibility*, according to humans, how well a highlight supports a predicted label [133, 29], and (ii) *faithfulness* or *fidelity*, how accurately a highlight represents the model's decision process [6, 127]. Human-annotated highlights (Table 2) are used to measure the plausiblity of model-produced highlights: the higher the overlap between the two, the more plausible model highlights are considered. On the other hand, a highlight that is both sufficient (implies the prediction, §2; first example in Table 2) and comprehensive (its complement in the input does *not* imply the prediction, §2; second example in Table 2) is regarded as faithful to the prediction it explains [29, 23]. Since human-annotated highlights are used only for evaluation of plausibility but not faithfulness, one might expect that the measurement and modeling of faithfulness cannot influence how human-authored explanations should be collected. In this section, we show that this expectation might lead to collecting highlights that are unfitting for the goals (ii) and (iii) in §1.

Typical instructions for collecting highlights encourage sufficiency and compactness, but not comprehensiveness. For example, DeYoung et al. [29] deem MOVIEREVIEWS and EVIDENCEINFERENCE highlights non-comprehensive. Carton et al. [23] expect that FEVER highlights are non-comprehensive, in contrast to DeYoung et al. [29]. Contrary to the characterization of both of these work, we observe that the E-SNLI authors collect non-comprehensive highlights, since they instruct annotators to highlight only words in the hypothesis (and not the premise) for neutral pairs, and consider contradiction/neutral explanations correct if at least one piece of evidence in the input is highlighted. Based on these discrepancies in characterization, we first conclude that post-hoc assessment of comprehensiveness from a general description of data collection is error-prone.

Alternatively, Carton et al. [23] empirically show that available human highlights are not necessarily sufficient nor comprehensive for predictions of *highly accurate* models. This suggests that the same might hold for gold labels, leading us to ask: are gold highlights in existing datasets flawed?

Let us first consider insufficiency. Highlighted input elements taken together have to reasonably indicate the label. Otherwise, a highlight is an invalid explanation. Consider two datasets whose sufficiency Carton et al. [23] found to be most concerning: neutral E-SNLI pairs and no-attack WIKIATTACK examples. Neutral E-SNLI cases are not justifiable by highlighting because they are obtained only as an intermediate step to collecting free-text explanations, and only free-text explanations truly justify a neutral label [20]. Table 2 shows one E-SNLI highlight that is not sufficient. No-attack WIKIATTACK examples are not explainable by highlighting because the absence of offensive content justifies the no-attack label, and this absence cannot be highlighted. We recommend (i) avoiding human-annotated highlights with low sufficiency when evaluating and collecting highlights, and (ii) assessing whether the true label can be explained by highlighting.

Consider a highlight that is non-comprehensive because it is redundant with its complement in the input (e.g., a word appears multiple times, but only one occurrence is highlighted). Highlighting only one occurrence of "great" is a valid justification, but quantifying faithfulness of this highlight is hard because the model might rightfully use the unhighlighted occurrence of "great" to make the prediction. Thus, comprehensiveness is modeled to make faithfulness evaluation feasible. Non-comprehensiveness of human highlights, however, hinders evaluating plausibility of comprehensive model highlights since model and human highlights do not match by design. To be able to evaluate both plausibility and faithfulness, we should annotate comprehensive human highlights. We summarize these observations in Figure 2 in Appendix A.

Mutual influence of data and modeling assumptions also affects free-text explanations. For example, the E-SNLI guidelines have far more constraints than the COS-E guidelines, such as requiring self-contained explanations. Wiegreffe et al. [128] show that such data collection decisions can influence modeling assumptions. This is not an issue per se, but we should be cautious that EXNLP data collection decisions do not popularize explanation properties as *universally necessary* when they are not, e.g., that free-text explanations should be understandable without the original input or that highlights should be comprehensive. We believe this could be avoided with better documentation, e.g., with additions to a standard datasheet [39]. Explainability fact sheets have been proposed for models [114], but not for datasets. For example, an E-SNLI datasheet could note that self-contained explanations were required during data collection, but that this is not a necessary property of a valid free-text explanation. A dataset with comprehensive highlights should emphasize that comprehensiveness is required to simplify faithfulness evaluation.

**Takeaways**

1. It is important to precisely report how explanations were collected, e.g., by giving access to the annotation interface, screenshotting it, or giving the annotation instructions verbatim.
2. Sufficiency is necessary for highlights, and EXNLP researchers should avoid human-annotated highlights with low sufficiency for evaluating and developing highlights.
3. Comprehensiveness isn't necessary for a valid highlight, it is a means to quantify faithfulness.
4. Non-comprehensive human-annotated highlights cannot be used to automatically evaluate plausibility of highlights that are constrained to be comprehensive. In this case, EXNLP researchers should collect and use comprehensive human-annotated highlights.
5. Researchers should not make (error-prone) post-hoc estimates of comprehensiveness of human-annotated highlights from datasets' general descriptions.
6. EXNLP researchers should be careful to not popularize their data collection decisions as universally necessary. We advocate for documenting all constraints on collected explanations

in a datasheet, highlighting whether each constraint is necessary for explanation to be valid or not, and noting how each constraint might affect modeling and evaluation.

## 5 Rise of Structured Explanations

The merit of free-text explanations is their expressivity, which can come at the costs of underspecification and inconsistency due to the difficulty of quality control (stressed by the creators of two popular free-text explanation datasets: E-SNLI and COS-E). In this section, we highlight and challenge one prior approach to overcoming these difficulties: discarding template-like free-text explanations.

We gather crowdsourcing guidelines for the above-mentioned datasets in Tables 6–7 in Appendix and compare them. We observe two notable similarities between the guidelines for the above-mentioned datasets. First, both asked annotators to first highlight input words and then formulate a free-text explanation from them, to control quality. Second, template-like explanations are discarded because they are deemed uninformative. The E-SNLI authors assembled a list of 56 templates (e.g., "There is ⟨hypothesis⟩") to identify explanations whose edit distance to one of the templates is <10. They re-annotate the detected template-like explanations (11% in the entire dataset). The COS-E authors discard sentences "⟨answer⟩ is the only option that is correct/obvious" (the only given example of a template). Template explanations concern researchers because they can result in artifact-like behaviors in certain modeling architectures. For example, a model which predicts a task output from a generated explanation can produce explanations that are plausible to a human user and give the impression of making label predictions on the basis of this explanation. However, it is possible that the model learns to ignore the semantics of the explanation and instead makes predictions based on the explanation's template type [66, 55]. In this case, the semantic interpretation of the explanation (that of a human reader) is not faithful (an accurate representation of the model's decision process).

Despite re-annotating, Camburu et al. [21] report that E-SNLI explanations still largely follow 28 label-specific templates (e.g., an entailment template "X is another form of Y") even after re-annotation. Similarly, Brahman et al. [18] report that models trained on gold E-SNLI explanations generate template-like explanations for the defeasible NLI task. These findings lead us to ask: what are the differences between templates considered uninformative and filtered out, and those identified by Camburu et al. [21], Brahman et al. [18] that remain after filtering? Are *all* template-like explanations uninformative?

Although prior work indicates that template-like explanations are undesirable, most recently, structured explanations have been intentionally collected (see Table 5; §3). What these studies share is that they acknowledge structure as *inherent* to explaining the tasks they investigate. Related work [GLUCOSE; 85] takes the matter further, arguing that explanations should not be entirely free-form. Following GLUCOSE, we recommend running pilot studies to explore how people define and generate explanations for a task *before* collecting free-text explanations for it. If they reveal that informative human explanations are naturally structured, incorporating the structure in the annotation scheme is useful since the structure is natural to explaining the task. This turned out to be the case with NLI; Camburu et al. [21] report: "Explanations in E-SNLI largely follow a set of label-specific templates. This is a *natural consequence of the task* and dataset". We recommend embracing the structure when possible, but also encourage creators of datasets with template-like explanations to highlight in a dataset datasheet (§4) that template structure can influence downstream modeling decisions. There is no all-encompassing definition of explanation, and researchers could consult domain experts or follow literature from other fields to define an appropriate explanation in a task-specific manner, such as in GLUCOSE [85]. For conceptualization of explanations in different fields see Tiddi et al. [119].

Finally, what if pilot studies do not reveal any obvious structure to human explanations of a task? Then we need to do our best to control the quality of free-text explanations because low dataset quality is a bottleneck to building high-quality models. COS-E is collected with notably less annotation constraints and quality controls than E-SNLI, and has annotation issues that some have deemed make the dataset unusable [87]; see examples in Table 7 of Appendix A. As exemplars of quality control, we point the reader to the annotation guidelines of VCR [143] in Table 8 and GLUCOSE [84]. In §6 and §7, we give further task-agnostic recommendations for collecting high-quality ExNLP datasets, applicable to all three explanation types.

**Takeaways**

1. EXNLP researchers should study how people define and generate explanations for the task before collecting free-text explanations.
2. If pilot studies show that explanations are naturally structured, embrace the structure.
3. There is no all-encompassing definition of explanation. Thus, EXNLP researchers could consult domain experts or follow literature from other fields to define an appropriate explanation form, and these matters should be open for debate on a given task.

## 6 Increasing Explanation Quality

When asked to write free-text sentences from scratch for a table-to-text annotation task outside EXNLP, Parikh et al. [92] note that crowdworkers produce "vanilla targets that lack [linguistic] variety". Lack of variety can result in annotation artifacts, which are prevalent in the popular SNLI [16] and MNLI [129] datasets [97, 45, 120], among others [40]. These authors demonstrate the harms of such artifacts: models can overfit to them, leading to both performance over-estimation and problematic generalization behaviors.

Artifacts can occur from poor-quality annotations and inattentive annotators, both of which have been on the rise on crowdsourcing platforms [25, 7, 87]. To mitigate artifacts, both increased **diversity of annotators** and **quality control** are needed. We focus on quality control here and diversity in §7.

### 6.1 A Two-Stage Collect-And-Edit Approach

While ad-hoc methods can improve quality [20, 143, 84], an effective and generalizable method is to collect annotations in two stages. A two-stage methodology has been applied by a small minority of EXNLP dataset papers [58, 144, 143], who first compile explanation candidates automatically or from crowdworkers, and secondly perform quality-control by having other crowdworkers assess the quality of the collected explanations (we term this COLLECT-AND-JUDGE). Judging improves the overall quality of the final dataset by removing low-quality instances, and additionally allows authors to release quality ratings for each instance.

Outside EXNLP, Parikh et al. [92] use an extended version of this approach (that we term COLLECT-AND-EDIT): they generate a noisy automatically-extracted dataset for the table-to-text generation task, and then ask annotators to edit the datapoints. Bowman et al. [17] use this approach to re-collect NLI hypotheses, and find, crucially, that having annotators edit rather than create hypotheses reduces artifacts in a subset of MNLI. In XAI, Kutlu et al. [67] collect highlight explanations for Web page ranking with annotator editing. We advocate expanding the COLLECT-AND-JUDGE approach for explanation collection to COLLECT-AND-EDIT. This has potential to increase linguistic diversity via multiple annotators per-instance, reduce individual annotator biases, and perform quality control. Through a case study of two multimodal free-text explanation datasets, we will demonstrate that collecting explanations automatically without human editing (or at least judging) can lead to artifacts.

E-SNLI-VE [32] and VQA-E [75] are two visual-textual datasets for entailment and question-answering, respectively. E-SNLI-VE combines annotations of two datasets: (i) SNLI-VE [131], collected by replacing the textual premises of SNLI [16] with FLICKR30K images [140], and (ii) E-SNLI [20], a dataset of crowdsourced explanations for SNLI. This procedure is possible because every SNLI premise was originally the caption of a FLICKR30K photo. However, since SNLI's hypotheses were collected from crowdworkers who did not see the original images, the photo replacement process results in a significant number of errors [122]. Do et al. [32] re-annotate labels and explanations for the neutral pairs in the validation and test sets of SNLI-VE. However, it has been argued that the dataset remains low-quality for training models due to artifacts in the entailment and the neutral class' training sets [78]. With a full EDIT approach, we expect that these artifacts would be significantly reduced, and the resulting dataset could have quality on-par with E-SNLI. Similarly, the VQA-E dataset [75] converts image captions from the VQA V2.0 dataset [43] into explanations, but a notably lower plausibility compared to a carefully-crowdsourced VCR explanations is reported in [78].

Both E-SNLI-VE and VQA-E present novel and cost-effective ways to produce large EXNLP datasets for new tasks, but also show the quality tradeoffs of automatic collection. Strategies such as crowdsourced judging and editing, even on a small subset, can reveal and mitigate such issues.

## 6.2 Teach and Test the Underlying Task

In order to both create and judge explanations, annotators must understand the underlying task and label-set well. In most cases, this necessitates teaching and testing the task. Prior work outside of ExNLP has noted the difficulty of scaling annotation to crowdworkers for complex linguistic tasks [106, 35, 99, 85]. To increase annotation quality, these works provide intensive training to crowdworkers, including personal feedback. Since label understanding is a prerequisite for explanation collection, task designers should consider relatively inexpensive strategies such as qualification tasks and checker questions. This need is correlated with the difficulty and domain-specificity of the task, as elaborated above.

Similarly, people cannot explain all tasks equally well and even after intensive training they might struggle to explain tasks such as deception detection and recidivism prediction [89]. Human explanations for such tasks might be limited in serving the three goals outlined in §1.

## 6.3 Addressing Ambiguity

Data collectors often collect explanations post-hoc, i.e., annotators are asked to explain labels assigned by a system or other annotators. The underlying assumption is that the explainer believes the assigned label to be correct or at least likely (there is no task ambiguity). However, this assumption has been shown to be inaccurate (among others) for relation extraction [8], natural language inference [96, 88], and complement coercion [35], and the extent to which it is true likely varies by task, instance, and annotator. If an annotator is uncertain about a label, their explanation may be at best a hypothesis and at worst a guess. HCI research encourages leaving room for ambiguity rather than forcing raters into binary decisions, which can result in poor or inaccurate labels [108].

To ensure explanations reflect human decisions as closely as possible, it is ideal to collect both labels and explanations from the same annotators. Given that this is not always possible, including a checker question to assess whether an explanation annotator agrees with a label is a good alternative.

**Takeaways**

1. Using a COLLECT-AND-EDIT method can reduce individual annotator biases, perform quality control, and potentially reduce dataset artifacts.
2. Teaching and testing the underlying task and addressing ambiguity can improve data quality.

## 7 Increasing Explanation Diversity

Beyond quality control, increasing annotation diversity is another task-agnostic means to mitigate artifacts and collect more representative data. We elaborate on suggestions from related work (inside and outside ExNLP) here.

### 7.1 Use a Large Set of Annotators

Collecting representative data entails ensuring that a handful of annotators do not dominate data collection. Outside ExNLP, Geva et al. [40] report that recruiting only a small pool of annotators (1 annotator per 100–1000 examples) allows models to overfit on annotator characteristics. Such small annotator pools exist in ExNLP—for instance, E-SNLI reports an average of 860 explanations written per worker. The occurrence of the incorrect explanation "rivers flow trough valleys" for 529 different instances in COS-E v1.11 is likely attributed to a single annotator. Al Kuwatly et al. [3] find that demographic attributes can predict annotation differences. Similarly, Davidson et al. [28], Sap et al. [110] show that annotators often consider African-American English writing to be disproportionately offensive.[2] A lack of annotator representation concerns ExNLP for three reasons: explanations depend on socio-cultural background [63], annotator traits should not be predictable [40], and the subjectivity of explaining leaves room for social bias to emerge.

On most platforms, annotators are not restricted to a specific number of instances. Verifying that no worker has annotated an excessively large portion of the dataset in addition to strategies from Geva

---

[2]In another related study, 82% of annotators reported their race as white [111]. This is a likely explanation for the disproportionate annotation.

et al. [40] can help mitigate annotator bias. More elaborate methods for increasing annotator diversity include collecting demographic attributes or modeling annotators as a graph [3, 126].

## 7.2 Multiple Annotations Per Instance

HCI research has long considered the ideal of crowdsourcing a single ground-truth as a "myth" that fails to account for the diversity of human thought and experience [9]. Similarly, EXNLP researchers should not assume there is always one correct explanation. Many of the assessments crowdworkers are asked to make when writing explanations are subjective in nature, and there are many different models of explanation based on a user's cognitive biases, social expectations, and socio-cultural background [82]. Prasad et al. [98] present a theoretical argument to illustrate that there are multiple ways to highlight input words to explain an annotated sentiment label. Camburu et al. [20] find a low inter-annotator BLEU score [91] between free-text explanations collected for E-SNLI test instances.

If a dataset contains only one explanation when multiple are plausible, a plausible model explanation can be penalized unfairly for not agreeing with it. We expect that modeling multiple explanations can also be a useful learning signal. Some existing datasets contain multiple explanations per instance (last column of Tables 3–5). Future EXNLP data collections should do the same if there is subjectivity in the task or diversity of correct explanations (which can be measured via inter-annotator agreement). If annotators exhibit low agreement between explanations deemed as plausible, this can reveal a diversity of correct explanations for the task, which should be considered in modeling and evaluation.

## 7.3 Get Ahead: Add Contrastive and Negative Explanations

The machine learning community has championed modeling *contrastive explanations* that justify why a prediction was made *instead of* another, to align more closely with human explanation [31, 49, 82]. Most recently, methods have been proposed in NLP to produce contrastive edits of the input as explanations [107, 134, 130, 55]. Outside of EXNLP, datasets with contrastive edits have been collected to assess and improve robustness of NLP models [59, 38, 74] and might be used for explainability too.

Just as highlights are not sufficiently intelligible for complex tasks, the same might hold for contrastive input edits. To the best of our knowledge, there is no dataset that contains contrastive free-text or structured explanations. These could take the form of (i) collecting explanations that answer the question "why...instead of...", or (ii) collecting explanations for other labels besides the gold label, to be used as an additional training signal. A related annotation paradigm is to collect *negative explanations*, i.e., explanations that are invalid for an (input, gold label) pair. Such examples can improve EXNLP models by providing supervision of what is *not* a correct explanation [112]. A human JUDGE or EDIT phase automatically gives negative explanations: the low-scoring instances (former) or instances pre-editing (latter) [58, 144].

**Takeaways**

1. To increase annotation diversity, a large set of annotators, multiple annotations per instance, and collecting explanations that are most useful to the needs of end-users are important.
2. Reporting inter-annotator agreement with plausibility of annotated explanations is useful to known whether there is a natural diversity of explanations for the task and should the diversity be considered for modeling and evaluation.

## 8   Conclusions

We have presented a review of existing datasets for EXNLP research, highlighted discrepancies in data collection that can have downstream modeling effects, and synthesized the literature both inside and outside EXNLP into a set of recommendations for future data collection.

We note that a majority of the work reviewed in this paper has originated in the last 1-2 years, indicating an explosion of interest in collecting datasets for EXNLP. We provide reflections for current and future data collectors in an effort to promote standardization and consistency. This paper also serves as a starting resource for newcomers to EXNLP, and, we hope, a starting point for further discussions.

## Acknowledgements

We are grateful to Yejin Choi, Peter Clark, Gabriel Ilharco, Alon Jacovi, Daniel Khashabi, Mark Riedl, Alexis Ross, and Noah Smith for valuable feedback.

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
