# OpenReview forum: "Teach Me to Explain: A Review of Datasets for Explainable Natural Language Processing"
_NeurIPS.cc/2021/Track/Datasets_and_Benchmarks/Round1 — NeurIPS 2021 Datasets and Benchmarks Track (Round 1)_

### Official Review · Reviewer_eyus · 2021-07-01
**Makes good points; could be better organized/written**

**Rating:** 9
**Confidence:** 4
**Correctness:** To my knowledge, the paper makes corr…

**Strengths:**

 - Makes novel points that are key to doing ExNLP properly
 - Provides constructive suggestions for improving current practices
 - Provides many starting points for future work on ExNLP

**Weaknesses:**

The paper is a little unfocused and comes across at some points as a collection of different points related to the same subject (datasets). It feels like some sections could be expanded into standalone papers in themselves. However, perhaps this is understandable in a paper that is tackling this subject (dataset quality in ExNLP) for the first time. I think more narrative connecting the different points would help the reader understand any ways in which the different points made contribute to a unified message.

**Additional Feedback:**

N/A

**Clarity:**

I found section 4 difficult to understand. In particular, the claim in Section 4 that human highlights are "independent" of the evaluation of faithfulness was difficult to understand without reading the whole section several times. It would help to explain:
- what do you mean by 'independence' here?
- who makes the independence assumption?
- what are the negative consequences of making the independence assumption?

I also found Table 5 in the appendix to be crucial to understanding this section. I think this table should be in the main text rather than the appendix.

**Documentation:**

Minor point: the authors don't explain how they arrived at the list of exNLP datasets. Were they taken from looking at all papers in a set of journals? Or just using their domain knowledge? We don't need an extensive description of this, but a couple of sentences describing the process would be good.

**Relation To Prior Work:**

Yes, it is clear that this is the first paper to question dataset annotation quality when it comes to ExNLP in particular. Other papers have started examining dataset quality in ML in general, but this is the first to focus on dataset quality in ExNLP.

**Summary And Contributions:**

This takes a much-needed lens to the details of how ExNLP datasets are actually collected, providing a corrective to the model-centric approach of much of machine learning. It points out some weaknesses in how we collect ExNLP data and how we document the collection process. The suggestions it makes for improvement are reasonable.

===Post author response===: Thanks for making the improvements. My rating is unchanged.

---

> ### Author Response · Authors · 2021-07-09
> **Response to Reviewer #3**
>
> Thank you for your positive and thoughtful feedback, and suggestions for improvement.  We have moved Table 5 from the Appendix into the main paper (page 2, Table 2) and added a discussion of our collection methodology in Appendix C.
>
> By "independent" we mean: since human-annotated highlights are used only for evaluation of plausibility but not faithfulness, it might seem that the measurement and modeling of faithfulness cannot influence how human-annotated explanations should be collected. This might lead to collecting highlights that are not suitable for the goals (ii) and (iii) in the Introduction. We included this clarification in the updated version of the paper above (see lines 145--148).

---

### Official Review · Reviewer_NJCs · 2021-07-02
**A comprehensive survey**

**Rating:** 7
**Confidence:** 3
**Correctness:** To my knowledge, the claims are solid.

**Strengths:**

A quite comprehensive and well-organised introduction and summary on large amounts of datasets for explainable NLP, which will be handy for both experts and beginners.

The suggestions for data collection provided can guide researchers for creating high quality data.


**Weaknesses:**

It is not very clear how to assess explanations in a dataset. Although the authors dedicate some sections for the related discussion, the discussion covers varies domains and dimensions, making it less focused. A short description in section 4 or 6 summing up key takeaways would be great. For instance, what qualifies a good or acceptable explanation? Is there a quantifiable measure?

**Additional Feedback:**

N/A

**Clarity:**

The paper is well written. However, it appears to me that section 4 to section 6 overall cover similar concept that is the data quality for EXNLP, making the perception that somehow the key messages are scattered here and there. It is a bit difficult to draw a high level picture from them.

**Documentation:**

Very detailed documentation.

**Relation To Prior Work:**

The distinctions between the submission and prior work are clearly made in section 1: the datasets are not addressed before and the authors want to connect the dots between the dataset and modeling experiments.

**Summary And Contributions:**

This survey paper provides a concise overview on 61 datasets for explainable natural language processing, highlighting their advantages/disadvantages and making suggestions for better data collection strategies. It provides a good starting point for EXNLP. The authors start with explaining why textual explanation is important for ML models, and then introduce three types of explanations and existing datasets. In particular they discuss how the dataset collection methods can affect modelling, focusing on the question of whether human explanation is sufficient or comprehensive for predictions (unsurprisingly, it is not). The paper further discuss means for improving the data quality and diversity, for instance, by using template-based explanations.

===Post author response===: Thanks for the responses clarifying the structure of the draft and incorporating the feedback in the final version. The final score remains unchanged as I already thought it can be published.

---

> ### Author Response · Authors · 2021-07-09
> **Response to Reviewer #2**
>
> Thank you for your thorough feedback. We integrated your suggestion to add main takeaways to the end of each section in the updated version of the paper posted above.
>
> Regarding what qualifies a good or acceptable explanation and how to quantify that. In this work, we accept all explanations that answer “why is [input] assigned [label]?” (Section 2). We highlight some necessary properties of human-annotated explanations (e.g., sufficiency) and conditions under which they are necessary (e.g., comprehensiveness if we wish to evaluate plausibility of model highlights that are constrained to be comprehensive), as well as properties that are previously typically considered as unwanted but we illustrate they are not necessarily bad (e.g., template-like explanations). However, there might be other good/bad properties of human-annotated explanations that we didn’t discuss since we focus on discussing topics related to the latest ExNLP and NLP research such as sufficiency, comprehensiveness, plausibility, faithfulness, template-like explanations, and data artifacts. Moreover, as we highlight in Section 5, there is no all-encompassing definition of explanation and thus there is no universal criteria for good explanations.
>
> Regarding the difference between Section 4–6. In Section 4 and Section 5, we aim to make two specific points, and we chose to highlight these two points because we judge them to be particularly important to the current ExNLP research. In particular, in Section 4, we aim to stress that evaluation and modeling of model-produced explanations influences how we collect human-annotated explanations, and other way around. For example, we should collect comprehensive highlights just because we cannot use non-comprehensive highlights to automatically evaluate plausibility of highlights that we constrained to be comprehensive (a typical modeling choice). In Section 5, we aim to highlight that template-like explanations are not necessarily bad as previously thought and illustrate that with E-SNLI. Then, in Section 6, we discuss how to control quality in explanation collection that can be beneficial regardless of modeling or evaluation choices.

---

### Official Review · Reviewer_dkRN · 2021-07-04

**Rating:** 6
**Confidence:** 3
**Correctness:** Correct to the best of my knowledge.

**Strengths:**

The paper is comprehensive and ambitious in scope. I found the paper useful as an overview of datasets with explanations in NLP, though admittedly as an “outsider” to the field so I cannot compare this meta-analysis with other meta-analyses of the same topic. I think this paper is a good resource for navigating the landscape of explainable NLP for outsiders. I found it especially helpful when the paper synthesizes or draws direct comparisons across datasets.

**Weaknesses:**

Overall the paper is a bit of a high-level "meta-analysis" summary, summarizing some meta-data of the datasets and pulling in/drawing out some specific attributes of specific datasets. I find the latter -- synthesizing discussions across datasets, comparing structural qualities across datasets, much more informative than the table by itself. The prescriptive guidelines and reflections in this paper are a bit high-level/descriptive and could be sharpened further. Although I could see this being a good resource for newcomers (Eg like a well-written blog post, for visibility); the sharpness of the analysis seems like there is room for improvement.

Although the taxonomy is based on formal structure of the explanation, e.g. splitting into highlight / free-form / structured, the paper remains fairly descriptive in summarizing or recounting the metadata of the datasets without finer-grained analysis. At times the editorial analysis (eg section 5) felt a bit informal or under-argued. While it was helpful to include specific examples, a more structured comparison of datasets along the evaluative criteria (eg sufficiency and so on for highlights) would have been helpful.

I also found it helpful when the paper recounted critiques of datasets, as this may be more evident to “insiders” to the literature than outside. Making this more structured (even including a column for possible concerns in the table --- though this could be  understandably a burden for the authors to maintain) could sharpen the utility of the summarizing tables.

**Additional Feedback:**


Minor writing comments:
Section 6.3: lines 300-303, it would have been helpful to flesh this out with more details; what kinds of tasks?
Lines 319-325: this argument seems a bit weak to me or self-evident, could somewhat be said more succinctly. What would be more effective than this paragraph, for example, would be including specific examples of the previously mentioned artifacts --- what should dataset creators watch out for specifically that could be introduced by not having enough annotators?

Typos: line 93, coarsest


**Clarity:**

The paper is reasonably well written although clarity lessens in the second half/last few sections.

**Documentation:**

This paper does not provide a novel dataset but rather conducts a "meta-analysis" of a category of datasets; its documentation is of the "metadata" of these datasets.

**Relation To Prior Work:**

It is somewhat discussed how this work differs from previous meta-analysis of explanations in ML and/or NLP.

**Summary And Contributions:**

This paper studies `"explainable NLP” and surveys available datasets and taxonomizes existing data collection methodologies. Focusing on a categorization of "highlights", "free-form", and "structured" explanations, the paper outlines how these categories appear in the datasets and certain weaknesses (eg artifacts introduced by annotators) or strengths that appear in certain datasets). There is also some high-level prescriptive guidance on best practices to carry forward.


====
Post author response:
Thanks to the authors for making the changes! I think they help on clarity.
I would keep the rating the same because as the authors mention some of the weaknesses are fairly inherent to the abridged survey form. I think the paper could be accepted.

---

> ### Author Response · Authors · 2021-07-09
> **Response to Reviewer #1**
>
> Thank you for your detailed feedback. Regarding the minor clarity issues you mentioned:  We have added more detail on tasks from prior work in Section 6.3 in the updated version of the paper above (lines 326-327). We have restructured Section 7.1 as suggested and focused on providing more examples of artifacts that dataset creators might look out for (see lines 346-357).
>
> Our analysis is in part high-level because of the relatively broad scope and the fact that we are the first to broach this area of explainable NLP datasets. We aimed for the survey to be comprehensive while still fitting into 9 pages so that it can serve the role of unifying an otherwise relatively disjoint research area (rather than a detailed analysis of a couple of datasets for a specific task, like the two most related prior works mentioned in lines 31-32). In theory, many of the subtopics could be expanded into more detailed papers of their own, but this wasn't our primary goal.

---

### Author Response · Authors · 2021-07-09
**Response to all Reviewers**

We thank the reviewers for their time and valuable feedback. We appreciate that all reviewers found the comprehensiveness and scope of our survey to be strong, found it to be a useful resource to newcomers & experts in explainable NLP, and rated in favor of acceptance! We note the point that this broadness of topics leads to some minor clarity issues between sections. Given the extra page, we have moved our "takeaways" section from the appendix to the main paper in order to improve clarity. We have updated the submission with the other style/writing suggestions given.

---

### Decision · Program_Chairs · 2021-07-26

**Decision:**

Accept

**Comment:**

The paper surveys existing datasets and data collection methodologies for explainable NLP, pointing out strengths and weaknesses of existing datasets and providing suggestions for future data collection in this area. Overall the reviewers found the paper interesting and useful to the community, but had some issues with the clarity and presentation. The authors were largely able to address these concerns with their responses and revised draft, and in the end all reviewers thought that the paper should be accepted. Congratulations on having your paper accepted to the NeurIPS 2021 Track on Datasets and Benchmarks! When preparing the camera-ready paper the authors are encouraged to take the reviewers feedback into account, specifically with improving the clarity and readability of the paper.